# Molecular Characterization and Pharmacology of Melatonin Receptors in Animals

**Erika Cecon** [1] , **Jean A. Boutin** [2,*] **and Ralf Jockers** [1]

1   Institut Cochin, Université Paris Cité, INSERM, CNRS, 75014 Paris, France
2   Laboratory of Neuronal and Neuroendocrine Differentiation and Communication, University of Normandy, INSERM U1239, 76000 Rouen, France
*   Correspondence: ja.boutin.pro@gmail.com

**Abstract:** Melatonin, the hormone of darkness, is secreted in minute amounts during the night and is virtually undetectable during the day. Melatonin mainly acts on high-affinity G protein-coupled receptors. The present review will trace the path of the discovery of melatonin receptors from their cloning, expression and purification to the development of recent radioactive and fluorescent tracers. We will then report on the state-of-the-art of melatonin receptor functional properties, including ligand bias and system bias due to receptor-associated proteins and receptor heteromers. Currently available antibodies raised against melatonin receptors will be critically reviewed here for the first time. The review will close with future perspectives in terms of the discovery of allosteric ligands and the in vivo validation of a range of melatonin receptor-associated signaling complexes to improve future drug development.

**Keywords:** melatonin; GPCR; molecular pharmacology; ligands; antibodies; functional pathways

## 1. Introduction

Melatonin is an indole-based natural compound derived from tryptophan and synthesized in the pineal gland by a well-described enzymatic cascade [1,2]. Once it became clear that melatonin was a key player in the regulation of circadian and circannual rhythms, it was not long until the binding site of melatonin responsible for these actions was identified as membrane-bound and mainly located in the hypothalamic–hypophysial brain region. Further studies determined two high-affinity binding sites corresponding to two distinct target proteins called $MT_1$ and $MT_2$, belonging to the G protein-coupled receptor (GPCR) superfamily [3]. A groundbreaking contribution in this respect was the design of an iodinated analogue of melatonin (2-[$^{125}$I]-iodomelatonin or 2-[$^{125}$I]-iodoMLT) [4] which not only remained a high-affinity agonist at those receptors [5,6], but also increased its potency (as usually happens with lipophilic substituents), allowing thus the discovery and localization of melatonin binding sites in different organisms and tissues.

A third high-affinity melatonin receptor was described in lower vertebrates such as xenopus [7], birds [8] and fishes [9]. This receptor, named Mel1c, was cloned and shown to have evolved in therian mammals into GPR50 based on a synteny of the *Mel1c/GPR50* genes and neighboring genes [10]. GPR50 lost its capacity to bind to melatonin but gained other roles in the regulation of melatonin receptors [11–13]. Evolutionary intermediate species permitted the tracing of this evolution using the platypus receptors [14]. Two other melatonin-binding sites were described over the years: *MT3* [15], which turned out to be the enzyme NQO2 [16,17], and the more recently described *MTx* [18] to which one can add a possible MT1d receptor proposed to exist in lower vertebrates [19,20] and a couple of low-affinity binding sites for which the physiological relevance remains to be demonstrated [21].

Systematic cloning and functional characterization of the main melatonin receptors in many different species helped to determine their pharmacological profiles, including

species-specific differences. The most recent splendid breakthrough was certainly the reports of the crystallizations of the human MT$_1$ and MT$_2$ receptors, including the description of the binding site topologies at the molecular level [22,23]. Those structures served as the basis for virtual docking campaigns [24,25]. Finally, the receptor structures (human MT$_1$ and MT$_2$) in complex with G$_i$ proteins and the melatonin receptor agonists ramelteon were obtained by cryo-electron microscopy [26,27] (Figure 1A,B). Generation of the 3D structures predicted for xMel1c and hGPR50 using AlphaFold [28,29] followed by their superposition with the experimentally determined active hMT$_1$ structure revealed the close resemblance of xMel1c and hGPR50 with hMT$_1$, in particular in the transmembrane region, supporting the reliability of the predicted structures (Figure 1C,D). The overlay also shows that the residue Q181 of hMT$_1$ that directly forms a hydrogen bond with the alkylamide side chains of melatonin receptor ligands is replaced by a tyrosine residue in the hGPR50. This is consistent with experimental observations showing that Q181 of the second extracellular loop is essential for melatonin binding in hMT$_1$, and its absence in hGPR50 likely contributes to its inability to bind melatonin [13]. The predicted structure of hGPR50 illustrates additional structural features of the long carboxyl-terminal tail of GPR50 such as the presence of an extended second alpha-helix separated by a proline residue from the canonical helix-8 present in most GPCRs. This predicted second helix is then followed by a long intrinsically disordered domain (Figure 1C).

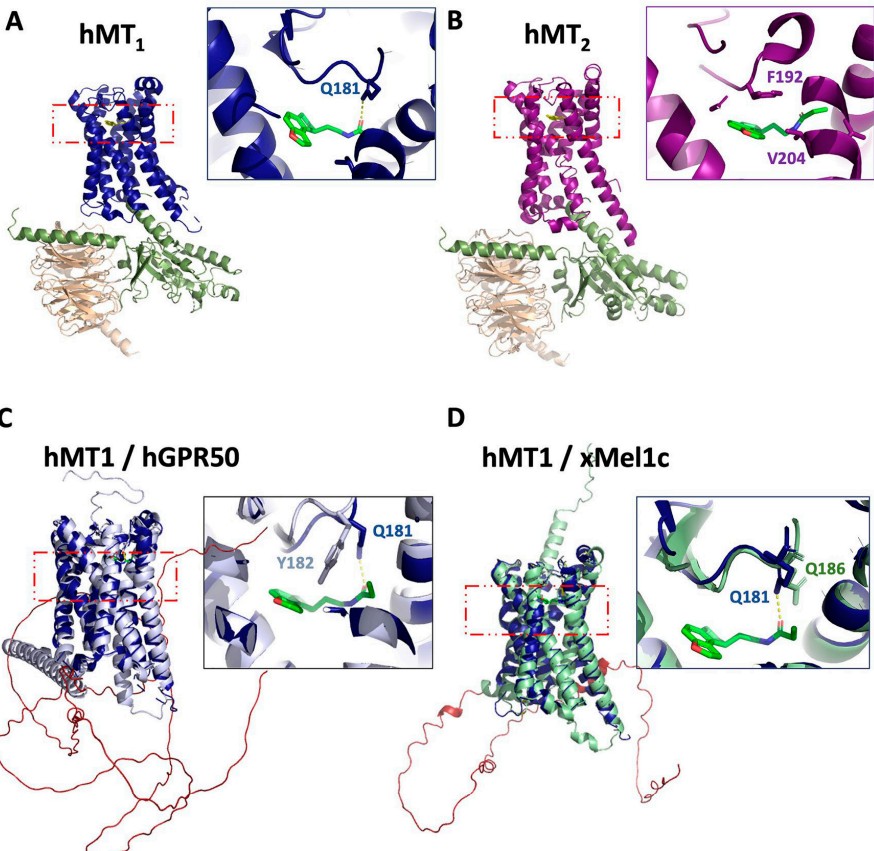

**Figure 1.** Molecular structures and models of melatonin receptors. (**A**) Human MT$_1$ (blue) (PDB 7DB6 [22]) and (**B**) MT$_2$ (violet) (PDB 7VH0 [23]) receptors bound to the agonist ramelteon and coupled to G$_i$ proteins (Gα (green); Gβ (ocher). AlphaFold [28,29] predicted structure of the human GPR50 (light blue) (**C**) and Xenopus Mel1c (pale green) (**D**) receptors superimposed with the human MT$_1$ structure (dark blue) (PDB 7DB6 [22]). Domains predicted to be intrinsically unstructured are shown in red. Insets show the ligand-binding sites occupied with ramelteon (green).

Herein, we aimed at summarizing the information on the animal melatonin field. Many progresses have been made on phytomelatonin, which have been nicely reviewed elsewhere [30–32].

The present review gives a short overview of the cloning, expression and purification history of these receptors followed by the description of radioactive or fluorescent tracers for these receptors, the functional properties of melatonin receptors, including biaism, the formation of molecular complexes containing melatonin receptor-associated proteins with distinct functional properties and the development of antibodies raised against these receptors. The review finishes by recapitulating available melatonin receptor drugs and future perspectives.

## 2. Cloning, Expression and Purification

### 2.1. Cloning

Historically, the cloning of the melatonin receptors commenced with the Mel1c receptor from Xenopus [7] followed by $MT_1$ and $MT_2$ from sheep and/or humans in the mid-1990s [33,34]. Although the cloning of human [33–35], rat (fragments [36]), mouse [35] and hamster [37] receptors were reported, their more complete molecular pharmacology started to be published only after 2004 for $MT_1$ and/or $MT_2$ receptors in human [38], rat [39], mouse [40], sheep [38,41] and hamster in part [42]; and Mel1c from platypus, hen and Xenopus [14]. Although the presence of the guinea-pig melatonin receptors has been recently claimed, neither $gpMT_1$ nor $gpMT_2$ were cloned and characterized. The authors describe similarities between iodomMLT-binding Kd in guinea pig, duck, chicken and human intestinal preparations and detected the receptors in small mucosa from guinea-pig using an anti-human melatonin receptor antibody. In the absence of cloning and full characterization of these receptors, as well as validation of the antibodies used against these targets, the existence of $gpMT_1$ or $gpMT_2$ awaits confirmation. [43,44]. Another laboratory species, rabbit, has also been widely used to characterize melatonin activity through its receptor(s). The only available data on melatonin receptors in this species seems to be the sequence (XM_017344057.1) predicted for $MT_2$ in rabbit (see review in Gautier et al. [45]). Table 1 summarizes the main characteristics currently available of some of those receptors across species and their affinity for a handful of ligands.

**Table 1.** Comparative pKis of melatonin receptor ligands across various species.

| | pKi | | | | | | | |
|---|---|---|---|---|---|---|---|---|
| | **Mouse MT1** | **Mouse MT2** | **Rat MT1** | **Rat MT2** | **Human MT1** | **Human MT2** | **Hamster MT1** | **Hen Mel1c** |
| **4P-PDOT** | 6.7 | 7.9 | 6.46 | 7.44 | 7.56 | 9.07 | 5.9 | 6.4 |
| **Luzindole** | 6.2 | 6.6 | 6.53 | 6.47 | 8.9 | 7.8 | 5.5 | 5.4 |
| **MLT** | 9.7 | 9.4 | 11.6 | 10 | 10.49 | 9.83 | 9.6 | 9.8 |
| **S24268** | 6.9 | 5.2 | 5.72 | 5.96 | 7.89 | 7.15 | N/D | N/D |
| **S20928** | 6.1 | 6.3 | 6.48 | 6.28 | 7.1 | 7.05 | 5.4 | N/D |
| **S24773** | 6.8 | 7.8 | 6.72 | 8.25 | 6.9 | 7.7 | N/D | N/D |

pKi values were determined in competition experiments with the 2-[$^{125}$I]-iodomelatonin tracer. N/D: not determined. Color code: Green compounds: agonists across a series of functional assays; Orange compounds: mostly antagonists, but can be partial agonists, depending on the used functional assay [46]. See also discussion in [47,48]. See Figure 2 for structures.

The difficulties in cloning melatonin receptor-expressing genes are nicely pinpointed by two species, sheep and hamster. For a long time, the melatonin community considered sheep as a natural knock-out of $MT_2$. After several years of experiments and some stubbornness, we finally succeeded in cloning the sheep $MT_2$ [41]. The reason why it was believed to be absent in the sheep genome seems to rely on the unexpected topology of the gene. Indeed, exon 1 and exon 2 are separated by about 10,000 base-pairs. Furthermore, that region is extremely rich in GC, rendering the cloning of this particular fragment extremely

tricky. By using a particularly stringent reverse transcriptase, Cogé et al. ended up passing this region and were able to clone the whole gene. This led to the expression of the sheep MT$_2$ in CHO cells followed by the characterization of its functional properties [41]. A similar situation occurred in the case of the hamster, one of the most studied species in melatonin physiology, particularly in terms of the reproductive cycle and hibernation [49]. Hamster genera are formed of *Mesocricetus* (e.g., Syrian or golden hamster), *Phodopus* (e.g., Djungarian hamster), *Cricetus* (e.g., European hamster), *Cricetulus* (e.g., Chinese hamster), *Allocricetulus* (e.g., Mongolian hamster), *Cansumys* (e.g., Gansu hamster) and *Tscheskia* (e.g., Korean hamster). Neumann et al. carefully studied their genetic relationship [50]. The Siberian hamster was clearly shown to be a natural knock-out for MT$_2$ at the genetic level [51], but the generalization of this fact to the other genera seemed questionable. Up to now, the very existence of the MT$_2$ gene in hamster species different from the Siberian hamster remains an open question [42].

Beyond these "standard" animals, melatonin receptor genes (or fragments of it) were cloned by us and others from a collection of different species that looked like a Noah's Ark: fishes, birds, mammals, insects, batracians, viper (*Daboia russelii*), bats (*Tadarida brasiliensis* and *Cynopterus titthaecheilus*), penguins (*Spheniscus humboldti* and *Aptenodytes patagonicus*), owl (*Strix uralensis*), hawk (*Accipiter nisus*), and eagle (*Aquila heliacal*) (see supplemental tables in Gautier et al. [45]).

A special mention should be added concerning the ever-elusive nuclear melatonin receptor. After an initial paper in 1994 claiming that the RZRβ nuclear receptor binds melatonin [52], a major discovery, it was necessary to confirm this finding independently. Many laboratories in the field of nuclear receptors attempted such a validation, unsuccessfully, until the initial paper was finally withdrawn [52]. Unfortunately, the initial publication is still cited about 550 times, mentioning that RZRβ is a nuclear receptor for melatonin. On the other hand, other laboratories started over again the search for a nuclear melatonin receptor. Up to now, still no convincing evidence has been gathered, even though some proteins such as HO1 and nrf2 are possible targets for the neurohormone [53–56] together with a possible interplay with retinoic acid [57].

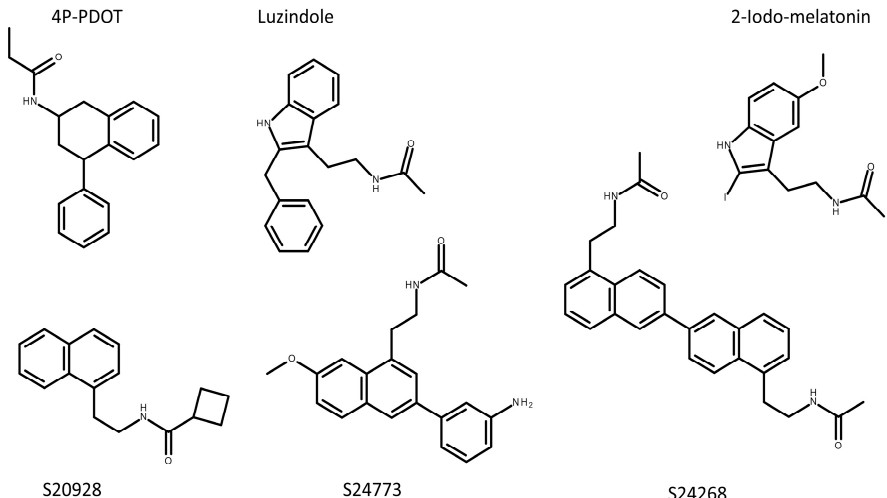

**Figure 2.** Structures of the ligands presented in Table 1.

### 2.2. Expression

Most of the genes of the previous section were only predicted based on whole genome sequencing efforts often representing only partial coding regions [45]. To increase the confidence of the identity of the gene in terms of the precise subtype, etc., the entire coding sequence should be isolated and sequenced before performing phylogenetic analysis and developing antibodies. The next logical step after cloning of a gene coding for a given

receptor is to express it in a cellular host—most of the time HEK293 or CHO cells. Expression and purification of GPCRs have been reviewed and include multiple procedures, the classical and more recent, alternative approaches [58]. Alternatives include microorganisms such as yeast or bacteria that express surprisingly high levels of proteins with some levels of functionality, while the membranes from those organisms are deeply different from mammals' as reviewed by Hartmann et al. [59]. High-level expression of melatonin receptors was mainly achieved in *Pichia pastoris* with a Bmax in the 500 pmol/mg Prot range [60]. Incidentally, attempts were made also with other expression systems: cell-free, *E. coli*, BHK cells (infected) and of course the standard HEK293 cells. The $MT_1$ receptor expression levels changed drastically between the cell-free system and *E. coli* (no expression) to BHK cells (2.4 pmol/mg), *P. pastoris* and stable HEK293 cells (>10 pmol/mg) [60].

In an attempt to demonstrate the action of melatonin on pathological situations through its $MT_1$ and/or $MT_2$ receptors, the receptors were expressed in unusual cell lines such as the 3T3 cell lines [61–63] for obesity or mesenchymal stem cells for osteogenesis [64].

## 2.3. Purification and Reconstitution

Purification of receptors has two purposes: the isolation of a single object within a defined milieu—such as nanodiscs or detergent micelles—for molecular, structural dynamics studies [60,65] and the supply of a pure object for crystallization [22,23]. The obvious difficulty for membrane proteins is that their three-dimensional structure is maintained by the phospholipid membrane forming the close and restrictive environment of the receptor [66]. Taking into account the fact that this restriction is synonymous of activity—if the structure changes slightly, the protein becomes inactive—it is quite important to substitute this lipid environment with a mixture of detergents, the nature and quantity of which is controlled, in order to find the mixture that maintains the activity of the protein. It is clear that this part is of an extremely difficult nature. Many alternative chemicals to lipid/phospholipids have been described during the last four decades, without a clear rationale for obtaining a comprehensive alternative to natural membranes [67,68]. Furthermore, the use of various detergents—the historical initial technique for solubilization/reconstitution—was complexified by the fact that the influence of different detergents led to different constraints on the protein, leading to modified specificities for integral membrane proteins (receptors and enzymes).

For the $MT_1$ receptor, expressed in *P. pastoris*, the solubilization was performed with 1% CHAPS or 0.25% Fos14. The receptor was then affinity-purified and reconstituted in a 0.1% Fos14/0.1% CHAPS mixture, leading to an active receptor with a Bmax of 1.5 nmol/mg Prot [60]. Alternatively, the receptor could also be reconstituted in nanoparticles, with one receptor per particle showing a pharmacological and functional profile similar to that obtained for receptor expressed in HEK293 cells [65]. This process followed a slightly different approach as membranes were solubilized with the SMA 2:1 copolymer, leading to characteristic lipid discs that sustained the receptor in a functional status [65].

## 3. Melatonin Receptor Ligands

### 3.1. Labeled Tracers for Melatonin Receptor

As stated previously, the successful synthesis of $^{125}$Iodo-melatonin was a key step in the domain of melatonin pharmacology. Binding is obviously the preferred technique to study receptors and to characterize them throughout the species and the compounds. It should be recalled here that finding alternative ligands—whether agonists or antagonists— is one of the key steps in the development of the arsenal to study receptors [1,69].

An extra difficulty resides in the high affinity of the receptors for their natural ligand, melatonin. Indeed, the pKd of tritiated melatonin for human $MT_1$ is 10.64 and for $MT_2$, 10.11 [70]. As a consequence, not only the ligands should be heavily radio-labelled, but also they should have high affinity at the receptor. This feature is quite exceptional in small molecule pharmacology, where it is not rare to find affinities of the main ligand at its receptor in the μM range such as for dopamine (pKd from 5 for $D_1$ and $D_2$ to 7 for

D$_3$ [71], serotonin (for example, its affinity is in the 10 μM range for 5-HT$_{5A}$ or 5-HT$_4$ receptors [72]) or histamine (pKi ~3.8 for H$_2$R [73] as reported in the IUPHAR database [www.guidetopharmacology.org/], although others reported pKi in the 6.6 range [74]).

In brief, three main labelled compounds are available to study melatonin receptors: [$^{125}$I-iodoMLT [4], the tritiated melatonin [70] and the *MT3*-targeting [$^{125}$I]-iodo-MCA-NAT [75], this last one presenting also a non-negligible affinity (IC50 in the μM range) for the MT$_2$ receptor [76]. Determination of binding affinities of these compounds for different targets is straightforward. Because these compounds are all agonist and are non-specific for either melatonin receptor subtype, we aimed for alternative compounds with high affinity that might show more distinct properties for MT$_1$ and MT$_2$ receptors. This resulted in the design of 2-(2-[(2-iodo-4,5-dimethoxyphenyl)methyl]-4,5-dimethoxy phenyl) (**DIV880**) and (2-iodo-*N*-2-[5-methoxy-2-(naphthalen-1-yl)-1*H*-pyrrolo[3,2-b]pyridine-3-yl])acetamide (**S70254**), two MT$_2$ specific ligands, and *N*-[2-(5-methoxy-1*H*-indol-3-yl)ethyl] iodoacetamide (**SD6**), an analog of 2-[$^{125}$I]-iodoMLT with slightly different characteristics [77,78]. A summary of their characteristics is presented in Table 2. The important part of this table is the non-MT$_1$-binding compounds S70254 and DIV880, which have a negligible affinity at MT$_1$. Indeed, several labeled ligands are typically needed to overcome existing limitations of a single molecule such as [$^{125}$I]-iodoMLT, which is unable to discriminate between MT$_1$ and MT$_2$ receptors. Labeled subtype-specific ligands are highly desirable to determine the specific receptor present in cells and tissues naturally expressing the receptor(s).

**Table 2.** Affinities of radioactive or fluorescent tracers at melatonin MT1 or MT2 receptors.

| Compound | pKd |
|---|---|
| **[$^3$H]-MLT** | |
| hMT$_1$ | 10.23 |
| hMT$_2$ | 9.46 |
| **2-[$^{125}$I]-MLT** | |
| hMT$_1$ | 10.69 |
| hMT$_2$ | 10.16 |
| **[$^{125}$I]-SD6** | |
| hMT$_1$ | 10.85 |
| hMT$_2$ | 10.18 |
| **[$^{125}$I]-S70254** | |
| hMT$_1$ | **No Aff** |
| hMT$_2$ | 9.61 |
| **[$^{125}$I]-DIV880** | |
| hMT$_1$ | **No Aff** |
| hMT$_2$ | 9.65 |
| **ICOA-9 (BODIPY-FL dye)** | |
| hMT$_1$ | 5.76 [a] |
| hMT$_2$ | 7.41 [a] |
| **ICOA-13 (BODIPY-FL dye)** | |
| hMT$_1$ | 4.97 [a] |
| hMT$_2$ | 5.48 [a] |
| **PBI-8192 (BODIPY NanoBRET 590 dye)** | |
| hMT$_1$ | **No Aff** [b] |
| hMT$_2$ | 7.26 [b] |
| **PBI-8238 (BODIPY NanoBRET 590 dye)** | |
| hMT$_1$ | 6.38 [b] |
| hMT$_2$ | **No Aff** [b] |

[a] pK$_i$ values from Gbahou et al. [79]. "**No Aff**" means no affinity. [b] pK$_d$ values from Gbahou et al. [80]. All other values are from Legros et al. [77].

Outside of the strict domain of labelled compounds, recent studies reported the development of fluorescent ligands at melatonin receptors [80] as well as the synthesis of

light-activatable caged melatonin compounds [81]. These molecules represent key elements in the toolbox for the pharmacological studies of melatonin receptors.

### 3.2. Virtual Melatonin Receptor Ligand Search

New pharmacophores with different structural chemical families are important in that they permit diversifying the tools with which one can study the receptors—in the molecular context—and move the studies to more physiological situations in which specificity and stability of the new compounds are essential. The available repertoire of compounds for melatonin receptors is still not as diverse from a chemical point of view as antagonists and agonists might belong to completely different chemical scaffolds, as is the case for the 5-HT GPCR superfamily (see Andrade et al. for details [82]). Indeed, for melatonin receptors, most of the 3000 agonists described were loose analogues of melatonin itself [48]. This, in fact, led to a rare situation in the GPCR landscape: receptors with "too many" agonists but barely a handful of antagonists, some of which were either very partial agonists or poor biased agonists [46]. New scaffolds might lead to molecules that are not complete antagonists but (very) partial agonists due to their biased pharmacology [46]. Particular attention should then be given to the functional characterization of these molecules (see next section). A remarkable analysis of the structural basis for melatonin receptor agonists has been recently published [83].

High-throughput docking techniques attempted to overcome this problem. Approximately, 150 million compounds were docked to the crystal coordinates in search of new pharmacophore(s) for melatonin receptors [84] with mixed results, and 62 virtually identified chemotypes were experimentally tested by Patel et al. [85]. Interestingly, some of those compounds [84] still belong to the general family of loose analogues of the melatonin's indole core while 15 new chemotypes were slightly different [24].

### 3.3. Functional Characterization of Melatonin Receptors

The functional and pharmacological characterization of melatonin receptors has historically relied on a few assays with a physiological readout in native tissues expressing endogenous melatonin receptors and on some biochemical assays in either endogenous or heterologous systems (i.e., cells expressing recombinant melatonin receptors). One of the first described physiological in vitro assays used for pharmacological characterization of melatonin effects takes us back to the times of the discovery of melatonin itself. It was in 1917 that McCord and Allen provided the first evidence of a pineal-derived factor able to alter the skin color of *Xenopus laevis* larvae [86]. Only in 1958 the active compound was isolated from bovine pineal glands by Lerner, who termed it "melatonin" in reference to its property to aggregate melanin granules, resulting in the previously observed skin-lightening effect [87]. Pigment aggregation assay in skin melanophores (pigment cells) became, thus, the first method employed to assess and characterize melatonin's effect. In fact, the very first suggestions of the existence of melatonin receptors, and of their mechanism of action, came from this assay in the late 1960s with the demonstration that melatonin was able to antagonize the forskolin effect on melanosome dispersion in Xenopus melanophores, and that this action was mediated through a "guanine nucleotide binding regulatory protein" (G protein) as it was blocked by pertussis toxin [88,89]. The assay has then also been used to test the potency of novel agonists and antagonists [90,91], and to compare melatonin's effect in additional models, such as fish and reptiles [92]. The detection of $MT_1$ and Mel1c receptors in the skin melanophores of *Xenopus L.* suggests that these are the molecular targets of melatonin mediating its effect on skin lightening [93]. The lack of selective ligands does not allow, for the moment, to discriminate the precise role and efficacy of each of these receptors in this experimental model.

The retina and the hypothalamic suprachiasmatic nucleus have also emerged as good models to functionally characterize melatonin and determine the structure activity relationship of melatonin and other indoles on the modulation of visual functions or of circadian rhythmicity, respectively [84,94–96]. After the cloning of melatonin receptors,

their heterologous expression helped to determine the function and pharmacological profile of each subtype with precision. By 1996, all cloned receptors were confirmed to be coupled to pertussis toxin-sensitive $G_i$ proteins, leading to inhibition of cAMP [3], as previously suggested by the functional assays. Heterologous expression systems also allowed the discovery of different signaling efficiencies for the cAMP and cGMP pathways of Xenopus Mel1c isoforms [97]. Among the biochemical assays still applied nowadays to characterize new ligands for melatonin receptors, cAMP is at the top of the list. In light of the important technological progress made in the past years, this technique is now much more straightforward and available in a non-radioactive format [98].

In addition to monitoring cAMP production, the assays of GTP-shift and [$^{35}$S]-GTP$\gamma$ binding have also been classically applied to characterize activation of melatonin receptors [46,99]. More recently, with the development of several Bioluminescence Resonance Energy Transfer (BRET)-based sensors for the investigation of GPCR functions [100], assessing several G protein subunits and isoforms' recruitment and activation or ß-arrestin recruitment became part of the repertoire of assays for melatonin receptor and ligands characterization [101,102]. Coupling to both G protein and ß-arrestin contributes to the formation of the high-affinity ternary complex comprising ligand–receptor–effector, typically revealed in the 2[$^{125}$I]-iodoMLT binding assays. Both melatonin receptors primarily couple to proteins of the $G_{i/o}$ subfamily, $MT_1$ couples in addition to $G_{12}$ and $G_{15}$ [103,104]. Both receptors recruit $\beta$-arrestin1/2 and activate the extracellular signal-regulated kinases 1/2 ($ERK_{1/2}$), and AKT [105]. Although both $MT_1$ and $MT_2$ receptors can activate ERK signaling at equal efficiencies, recent evidence suggests that they differ in the upstreaming signaling molecules and an interesting cooperativity between $G_i$ and $G_q$ proteins has been shown to exist in the case of $MT_2$ [106]. Additional activation of the phospholipase C/$Ca^{2+}$ pathway by $MT_1$ has been observed in several experimental systems by several laboratories. In some cellular systems, this effect was indirect, through $G\beta\gamma$ liberated upon activation of the pertussis toxin-sensitive $G_i$ protein [107], while in other systems the direct activation occurs by the pertussis toxin-insensitive $G_{q/11}$ protein [108]. An overall view of the main signaling pathways triggered by melatonin upon activation of $MT_1$ and $MT_2$ is shown in Figure 3.

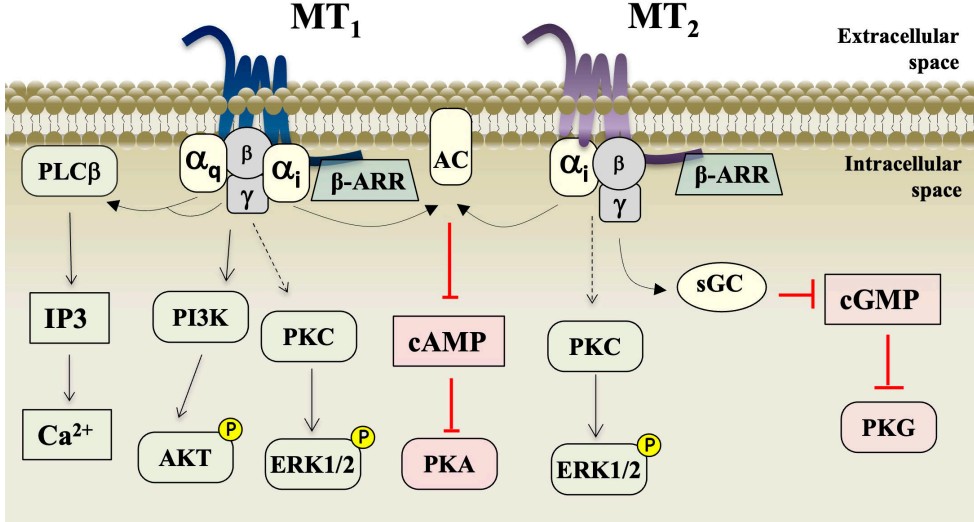

**Figure 3.** Melatonin receptor signalling pathways. Melatonin activation of MT1 and $MT_2$ receptors triggers mainly G$\alpha$i activation, decreasing cAMP levels and activation of the MAP Kinases cascade, resulting in increased levels of phospho-ERK. Both receptors can also recruit ß-arrestins (ß-ARR). MT1 also activates PI3K/Akt and PLC/$Ca^{2+}$ pathways in.a G$\beta\gamma$-dependent or G$\beta\gamma$-dependent manner, while activation of MT2 receptors inhibits cGMP levels. See text for details. $\beta$-ARR, $\beta$-arrestin; AC: adenylyl cyclase; sGC, soluble guanylyl cyclase.

*3.4. Bias*

For several GPCRs, a more in-depth characterization of their functional pharmacology revealed that some ligands display preferential functionality on specific signaling pathways to the detriment of other pathways, a behavior that defines them as "biased ligands" (see [46,109,110] for details). Such characterization has proven to be of extreme value for drug development strategies, allowing the selection of ligands more suitable for therapeutic applications that could exclusively activate a desired signaling pathway over other pathways potentially linked to adverse effects. For instance, a recent genetic and functional investigation of $MC_4$ receptors, a major drug target to treat obesity, indicated that biased signaling towards β-arrestin 2 recruitment over $G_{\alpha s}$ signaling is protective against obesity, thereby opening great perspectives for the development of ß-arrestin-biased agonists [111]. In the case of melatonin receptors, this aspect is still at its infancy for several reasons. Firstly, the precise signaling pathway mediating a specific melatonin effect is known in only a few cases. For instance, the synchronization effect of melatonin acting on $MT_1$ and $MT_2$ receptors in the hypothalamic suprachiasmatic nucleus is associated to the activation of the $G_i$ pathway, although different downstream signaling molecules are involved for each receptor [112,113] Another example of a better characterized effect of melatonin is in retinal functions, where melatonin modulates the electrical activity of the retina in response to light stimuli under low-light conditions through the PKCζ pathway [114]. The second reason why the ligand bias aspect is still poorly explored in the melatonin receptor field is the fact that only few studies attempted to characterize the full properties of ligands with regard to several signaling pathways. The first melatonin receptor ligand defined as a biased ligand was a recently developed fluorescent agonist, ICOA-9, which showed preferred signaling through the $G_i$/cAMP pathway over β-arrestin or MAPK/$ERK_{1/2}$ pathways [79,115]. More recently, a first real attempt to find biased ligands at melatonin receptors among the several existing synthetic ligands was performed [46]. In this major effort, 21 melatonergic ligands were functionally characterized in five different assays: GTPγS binding, cAMP production, internalization, β-arrestin recruitment and cell morphology changes (CellKey®). This study revealed that some ligands classically known as antagonists, including the two golden-standard reference ligands luzindole and 4-P-PDOT for the cAMP pathway, can display a partial agonistic activity in other signaling pathways. Thus, these data reinforce (i) the importance and urgent need for a better characterization of existing and newly described melatonergic ligands, and (ii) the necessity for a more in-depth investigation of the melatonin signaling pathways underlying each effect in order to allow their potential therapeutic application. Finally, the clinically available drug agomelatine, prescribed for its antidepressant effects (see next section for further details), can also be considered a biased ligand, as it targets the heterodimer formed of $MT_2$ and the serotonin receptor 5-$HT_{2c}$. This heterodimer is a different signaling unity compared to each receptor on its monomeric form and whose activation by agomelatine seems to privilege the $G_i$/cAMP over the $G_q$/PLC pathway, which is actually antagonized [115,116].

*3.5. Receptor-Associated Proteins*

We tend to characterize the function of GPCRs and their ligands by expressing them in fibroblastic standard cells such as HEK293 or CHO cells which may or may not overexpress, in addition, primary signal transducers such as specific G proteins, β-arrestins or GPCR-regulated kinases (GRK). By following this approach, the community generated standardized G protein and β-arrestin coupling profiles for each receptor that are comparable between different receptors [117]. For human $MT_1$ and $MT_2$ receptors, these coupling profiles have been determined in BRET assays in transfected HEK293 cells [103,104,117]. Both receptors couple to β-arrestin1/2 and all members of the $G_{i/o}$ family including $G_z$ in these assays. For $MT_1$, additional coupling to $G_{12}$ and $G_{15}$ was observed [103,104]. When the coupling profile is determined for endogenously expressed G proteins, the result is slightly different as it has been shown in co-immunoprecipitation experiments in HEK293 cells receptors [97]. Whereas $G_{i2}$ and $G_{i3}$ proteins were present in precipitates, $G_{i1}$, $G_o$, $G_z$ and

$G_{12}$ proteins were absent despite detectable expression levels in HEK293 cells. Coupling to $G_{i2/3}$ was particularly strong and detectable even in the absence of agonists, leading to a measurable constitutive activity of $MT_1$ at low expression levels [107]. Surprisingly, $G_{q/11}$ proteins were also co-precipitated, an observation that was consistent with the transient elevation in cytosolic calcium concentrations by melatonin in these cells [108]. Altogether, standardized G protein and β-arrestin coupling assays allow defining the repertoire of possible couplings, but the situation might be quite different in a given cellular context expressing a specific repertoire of signal transducers at endogenous expression levels.

The second series of studies that helped to identify melatonin receptor-associated proteins on a broader scale is based either on proteomic or genetic approaches that aim to identify associated proteins without previous knowledge of their identity [118]. By combining the results of these approaches, a virtual $MT_1/MT_2$ network was defined that contained 366 interacting partners of which 168, 143, and 52 were specific for $MT_1$, $MT_2$ or common for both receptors, respectively [119]. Among them, a subnetwork possibly present in synapsis has been identified in which Munc-18, the voltage-gated calcium channel CaV2.2, SNAP25, synapsins, n-NOS, PSD-95 and the dopamine transporter (DAT) have been validated by further experiments [119–121].

Melatonin receptor-associated proteins can be divided into three classes: GPCRs forming receptor heteromers, other transmembrane proteins and intracellular proteins. For $MT_1$, two cytosolic proteins have been described to physically interact with the C-terminal tail to regulate downstream signaling. The first one is the regulator of G protein signaling (RGS)20 that binds at the level of helix 8 of the proximal part of the $MT_1$ C-terminus [120]. RGS proteins are known to directly interact and modulate the function of G proteins [122] but can, in some cases, also directly interact with receptors [123–125]. In the context of $MT_1$, RGS20 accelerates the activation kinetics of melatonin-induced G protein and KIR3 channel activation [120,126]. Molecular proximity BRET studies indicate an important role of $MT_1$ homodimers with one protomer interaction with RGS20 and the other with the $G_i$ protein [126]. RGS20 is likely to modulate $MT_1$ function in ovine pituitary *pars tuberalis*, a major site of $MT_1$ expression involved in seasonal reproduction [120]. The second cytosolic-associated protein is the multi-PDZ domain protein MUPP1, which interacts with its PDZ10 domain and the PDZ domain-binding motif 'DSV' of $MT_1$ corresponding to its last three amino acids [127]. Disruption of the $MT_1/MUPP_1$ interaction showed that MUPP1 destabilizes the interaction of the $G_i$ protein with $MT_1$ and is necessary for $G_i$-dependent downstream signaling through the adenylyl cyclase and ERK pathways. The formation of the $MT_1/MUPP1$ complex has been shown in ovine pituitary *pars tuberalis* but is likely to operate in many other cell types as MUPP1 is ubiquitously expressed [127].

Concerning melatonin receptor heterodimers, the first example concerns the formation of heteromers between $MT_1$ and $MT_2$ [128,129] that seem to be particularly important in the retina, where they are coupled to the activation of the $G_q/PLC/PKC_\xi$ pathway to increase the night-time light sensitivity of the retina [120,130]. Another heteromer was discovered between $MT_2$ and the serotonin $5-HT_{2c}$ receptor, which seems to be the predominant $MT_2$ complex in comparison to $MT_2$ homomers, in the mouse brain [115,131]. Within this heteromer, melatonin and the clinically used antidepressant agomelatine activate the $G_q/PLC$ pathway by transactivating the $G_q$-coupled $5-HT_{2c}$ receptor. Heteromerization confers new functional properties as classical melatonin receptor antagonists for the $G_i/AC$ pathway, such as 4-PPDOT and luzindole, are agonists for the $G_q/PLC$ pathway. Further described heteromers are those between $MT_1$ and GPR50 in which GPR50 behaves as a negative allosteric modulator of $MT_1$ ligand binding and signal transduction [11], and between $MT_2$ and various orphan (GPR61, GPR62, and GPR135) receptors [132]. The expression pattern and the physiological relevance of these complexes remain to be demonstrated. For a more detailed description of melatonin receptor heteromers, the reader is invited to consult a recently published review [116].

Melatonin receptors have been shown to interact with other transmembrane proteins. $MT_1$ and $MT_2$ form a complex with DAT in mouse striatal synaptosomes and

inhibit the capacity of dopamine re-uptake through DAT [121]. Consistently, a decreased amphetamine-induced locomotor activity was observed in melatonin receptor knock-out mice. In vitro studies indicate that melatonin receptors retain DAT in the endoplasmic reticulum in its immature non-glycosylated form, thereby regulating the amount of DAT available at the cell surface.

$MT_1$ was shown to form a complex with the voltage-gated calcium channel Cav2.2 [119]. In vitro studies indicate that $MT_1$ tonically inhibits $Ca^{2+}$ current entry which can be further amplified by $MT_1$ activation, presumably mediated by $G_{\beta\gamma}$ rather than a voltage-independent pathway. The physiological significance of this observation remains to be shown but a similar mechanism has been previous observed for opioid receptors [133,134].

The heteromer between the orphan GPR50 and the TβRI receptor responding to the transforming growth factor-β (TGFβ) is an interesting case as it renders the TβRI constitutively active by stabilizing the active TβRI conformation and by displacing the negative regulator $FKBP_{12}$ from TβRI [12]. These results show the existence of an alternative molecular complex in which GPR50 replaces the TβRII that is responsible for TGFβ binding in the canonical TβRI/TβRII complex. TGFβ signaling is known to be antiproliferative and inhibitory on mammary tumor development at early stage of tumorigenesis [135]. In vivo studies revealed the anticipated protective effect of GPR50 against cancer development, as targeted deletion of GPR50 in the MMTV/Neu spontaneous mammary cancer model showed decreased survival after tumor onset and increased tumor growth [12].

In conclusion, melatonin receptor-associated proteins can have profound effects on their pharmacological profile and functions and in turn can also have important modulatory functions on their associated proteins. These examples illustrate the impact of system bias on receptor function and the necessity to study melatonin receptor function in physiologically relevant cellular contexts.

*3.6. Antibodies*

Antibodies are indispensable tools to monitor the expression of proteins in cells and tissues. More recently, antibodies and single-domain antibody fragments (also known as VHH or nanobodies), are increasingly used as therapeutic or research tools to modulate the function of proteins, in particular GPCRs [136].

The generation of antibodies directed against melatonin receptors has proven to be difficult. Like most other members of the class A GPCR family, these transmembrane-spanning receptors expose only a limited surface that can be recognized by antibodies. There exist essentially two regions, one exposed to the intracellular and the other to the extracellular site. Each is composed of three short loops and a relatively short N-terminal or C-terminal domain (see Figure 1 for receptor structures). The long C-terminal tail of mammalian GPR50 is a notable exception.

Many attempts have been made over the years by us and others to obtain antibodies recognizing the three members of the melatonin receptor subfamily, in particular the human and rodent proteins. Antibodies can be found in the catalogue of all specialized antibody providers. However, key information on the specificity of these antibodies is typically missing. Often, Western blots or immune-histological images of endogenously expressed receptors are shown, but without adequate controls such as knock-out cells/tissues, thus the specificity of these signals remains questionable. In some cases, the specificity issue has been addressed by pre-incubating the antibody with an excess of immunogenic peptide, which is however not sufficient proof as the 'non-specific' antigens are also expected to be displaced by the immunogenic peptide.

The determination of the antibody specificity often needs to be addressed in a dedicated study, either by comparing cells expressing the recombinant receptors and those not expressing them or by comparing cells/tissues expressing endogenous receptors with cells/tissues devoid of them (typically from knock-out cells or tissues). For $MT_1$ and $MT_2$ receptors, up to now, antibodies from six different sources have been validated for each receptor following this specificity check. The first generation of polyclonal antibodies

includes the 536-antibody raised by our laboratory against the last 19 amino acids of the human $MT_1$ [108]. This antibody specifically recognizes the recombinant and endogenously expressed human $MT_1$ in various human tissues [108,137–147], but neither the human $MT_2$ nor the mouse, rat, or hamster $MT_1$ (personal communication RJ). At the same time, the Fraschini laboratory generated polyclonal antibodies against the last nine amino acids of the human $MT_1$ and $MT_2$ [148]. In particular, the anti-$MT_2$ antibody has been validated with the recombinant receptor and has recognized endogenously expressed receptors in various human tissues [137–139,147–149]. Both antibodies cross-react with rat but not with mouse receptors and have been used to map receptor expression in rat brain [150,151].

The Tosini laboratory generated a polyclonal antibody raised against the last 19 amino acids of the mouse $MT_1$ [152]. The antibody was validated in retinal extracts of wild-type and $MT_1$ knock-out mice using Western blot and immunostaining experiments. The Stehle laboratory characterized the specificity of commercially available anti-$MT_1$ and anti-$MT_2$ antibodies in the mouse hippocampus [153]. Polyclonal antibodies commercialized by Santa Cruz (#SC13186, #SC13187) and raised in goat against melatonin receptors showed an immunoreactive band at the expected seize in Western blot of hippocampal extracts and an immunoreactivity in brain slices that was absent in $MT_1/MT_2$ double knock-out mice. A similar labeling pattern was observed in hippocampal brain slices with two antibodies commercialized by Alomone (#AMR-031, #AMR-032) and the anti-$MT_1$ antibody described by Sengupta et al. [152]. The specificity of the hippocampal signal obtained with the Sengupta antibody was confirmed in $MT_1$ knock-out mice. The specificity of the anti-$MT_2$ antibody from Santa Cruz was further validated in the cone-like photoreceptor cell line 661W that express endogenous melatonin receptors [154]. Knock-out of $MT_2$ in this cell line abolished the signal in immunofluorescence experiments and in the proximity ligation assay aiming to detect $MT_1/MT_2$ heteromers in combination with the anti-$MT_1$ antibody from Alomone. Anti-$MT_2$ antibodies from Alomone showed a similar immunoreactivity immunofluorescence experiments in 661W wild-type cells [155] and detected neurons in the periaqueductal gray in mice [156].

More recently, several monoclonal antibodies have been generated against the entire C-terminus of the mouse $MT_1$ and $MT_2$ sequences [157]. Several antibodies recognized either recombinant mouse $MT_1$ or $MT_2$ receptors in Western blot, immunoprecipitation and co-immunoprecipitation experiments. Some antibodies cross-reacted with human or rat receptors. Endogenous receptors were detected as melatonin receptors at the expected locations in mouse such as the retina, suprachiasmatic nuclei, pituitary gland [157] and striatal synaptosomes [121]. Specificity was demonstrated using tissues from melatonin receptor knock-out mice. These antibodies detected also $MT_1$ and $MT_2$ receptors in the glomerular layer of the rat olfactory bulb [158].

For the orphan GPR50, polyclonal antibodies were successfully raised against the last 13 amino acids of the human and ovine receptors [159,160]. These antibodies recognized human, mouse, rat and ovine GPR50. Endogenously expressed receptors were detected in the suspected locations, such as the *dorsomedial nucleus* of the hypothalamus, tanycytes of the *median eminence* and the pituitary in all the three species studied. GPR50-expressing cells were also observed in several other regions such as the amygdala, sub-fornical organs and the hippocampus [159–161]. (See list of validated anti-melatonin receptor antibodies shown in Table 3.)

**Table 3.** Validated antibodies recognizing receptors of the melatonin receptor family.

| Antibody Target | Host | Epitope | Antibody Source | IHC/IF | WB | IP | Reported Cross-Reactivity | Validation |
|---|---|---|---|---|---|---|---|---|
| MT$_1$ | rabbit | last 19 aa hMT$_1$ | Brydon et al. [97] | 1:100 | 5 µg/mL | 1:40 | h (not m, r, ham) | transfected cells |
| | rabbit | last 9 aa hMT$_1$ | Angeloni et al. [148] | 1:100 | — | — | h | transfected cells |
| | rabbit | last 19 aa mMT$_1$ | Sengupta et al. [152] | 1:500 | 1:500 | — | m | KO mice tissue |
| | goat | not provided | Santa Cruz #SC13186 | 1:200 | 1:500 | — | m | KO mice tissue |
| | rabbit | residues 223–236 of ICL3 | Alomone #AMR-031 | 1:500 | 1:200 | — | m, (h, r) [a] | similar IF staining pattern [b] |
| | mouse | entire Cter mMT$_1$ | Cecon et al. [98] | 10 µg/mL | 2 µg/mL | 1 µg/mL | m (h, r for some antibodies) | transfected cells, KO mice tissue |
| MT$_2$ | rabbit | last 9 aa hMT$_1$ | Angeloni et al. [148] | 1:1000 | 1:500 | 1:500 | h, r (not m) | transfected cells, KO mice tissue |
| | goat | not provided | Santa Cruz #SC13177 | 1:200 | 1:500 | — | m | KO mice tissue, KO 661W cell line |
| | rabbit | residues 232–246 of ICL3 | Alomone #AMR-032 | 1:250 | 1:2000 | — | m, (r) [a] | similar IF staining pattern [c] |
| | mouse | entire Cter mMT$_1$ | Cecon et al. [98] | 10 µg/mL | 2 µg/mL | 1 µg/mL | m (h, r for some antibodies) | transfected cells, KO mice tissue |
| GPR50 | rabbit | last 13 aa hGPR50, oGPR50 | Hamouda et al. [159] | 1:150–1:2000 | 1:1000 | 4–10 µg/mL | h, m, o, r | transfected cells, KO mice tissue |

[a] according to the provider. [b] similar IF staining pattern as Santa Cruz #SC13186 and Sengupta et al. antibodies in mouse hippocampal brain slices. [c] similar IF staining pattern as Santa Cruz #SC13177 in IF experiments in 661W wild-type cells. aa, amino acid; Cter, carboxyl terminal tail; ham, European hamster; h, human; ICL3, intracellular loop 3; IF, immunofluorescence, IHC, immunohistochemistry; IP, immunoprecipitation; KO, knock-out; m, mouse; o, ovine; r, rat, WB, Western blot.

## 4. Drugs at Melatonin Receptors

Melatonin was very early (1964) used in sleeping situations [162]. Despite the fuzziness of melatonin actions, several pharmaceutical companies conducted intensive research and findings of melatonin receptor agonists leading to interesting compounds on the market for the minor condition of jetlag and insomnia (Ramelteon® [163]) and the surprisingly good behavior of a melatonin receptor agonist in depression [164] (agomelatine, Valdoxan®). Those studies were conducted according to a strictly validated protocol in several clinical studies in different countries [165].

In addition to anti-depressant effects, agomelatine (Valdoxan) has demonstrated some efficacy in treating generalized anxiety disorder (GAD) [166] in the most recent study, after two decades of clinical evidence of the efficacy of this compound in clinical studies to treat depression [167–169].

Tasimelteon (Hetlioz), another melatonin receptor agonist, has been described as a major compound in treating circadian rhythm disorder, particularly in blind people [170].

All these compounds are unable to discriminate between MT$_1$ and MT$_2$ receptors, which somehow limits its precision for some applications such as sleep regulation, for which different and partially opposing effects are suggested for MT$_1$ and MT$_2$ receptors in rodents [171].

Finally, several pharmaceutical companies have developed time-released melatonin tablets, a compound that proved to ameliorate the sleep quality of elderly patients [172] and autistic children [173].

## 5. Future Trends and Conclusions

As time passes by, the landscape of GPCR functions seems to have been populated with one more level of complexity. Two of these levels should be more delineated in the following years. The first is the discovery of natural ligand(s) acting as allosteric modulators, such as has been suspected recently for the formidable complexity of the sweet taste receptors [174]. These compounds would be of great interest, also from a therapeutic point of view, as they might modulate the specificity, affinity and potency of the reference ligands including the natural ligand, melatonin in our case. Whether these initial observations on the taste GPCRs will be a generality for all receptors remains to be

shown. It must be pointed out that these discoveries are variations of the theme of the many studies reporting biased signaling [175]. The second source of complexity is the role of ions as endogenous allosteric modulators of GPCRs, as reviewed by van der Westhuizen et al. [176]. This has a direct impact, if validated also for melatonin receptors, on our descriptions of agonists as well as our understanding of the behavior of the receptors in physiological systems.

The recent descriptions of dimeric/oligomeric complexes for several GPCRs from different families also have to be further validated in a more natural context than the majority of approaches using transfected cells. Demonstration of receptor dimers in expression systems, as we did in the case of the $MT_2/5-HT_{2c}$ complex [115], is found only in a minority of reports. This very point is also the biggest challenge for the years to come in receptor pharmacology: how many of our data observed in artificial systems (host/transgenes) will be validated in situ? And how? Indeed, this formidable challenge will necessitate several technical ruptures in our experimental designs, particularly concerning our capacity to develop models closer to the physiology, probably involving stem cell-derived cell lines with or without the help of possibly personalized stem cell recovery/isolation approaches. In this context, the development of 'biologicals' such as high-affinity antibodies or VHH [177] as pharmacological tools or therapeutic agents are expected to make major contributions to solve some of the mysteries linked to the presence—or not—of melatonin receptors in various organs.

Finally, further understanding of the relationship between melatonin receptor signaling and melatonin effects will allow better defining the role/influence of receptor-associated proteins and fuel the rational design of signaling-pathway-biased ligands linking to specific signaling pathways with desirable effects in the years to come.

**Funding:** Agence Nationale de la Recherche: ANR-19-CE16-0025-01 «mitoGPCR» and ANR-21-CE18-0023 «alloGLP1R»; ANR: Appel RA-Covid-19 projet MELATOVID (ANR-20-COV4-0001); Fondation pour la recherche Médicale: Equipe FRM DEQ20130326503. La Ligue Contre le Cancer N/Ref: RS19/75-127 and the Institut National de la Santé et de la Recherche Médicale (INSERM), Centre National de la Recherche Scientifique (CNRS).

**Institutional Review Board Statement:** Not applicable.

**Informed Consent Statement:** Not applicable.

**Data Availability Statement:** Not applicable.

**Conflicts of Interest:** The authors declare no conflict of interest.

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
