# Peer review of "Molecular Characterization and Pharmacology of Melatonin Receptors in Animals"

_2813-2564, doi:10.3390/receptors2020008_

Round 1
Reviewer 1 Report
The present review presents the path of the discovery of melatonin receptors from its cloning, expression and purification to the development of radioactive and fluorescent tracers. The study will enhance our understanding of melatonin production and pathways. However, there are some deficiencies which must be addressed.
In the abstract main findings should be added.
Conclusion of the study and future recommendation should be provided in the abstract.
Line 35-36 “Further studies” should be cited with relevant studies.
Line 116 lack reference and should be cited with relevant studies. The following studies may be helpful.
https://doi.org/10.3390/genes13101699, https://doi.org/10.3390/life12111922,
scheme or pathway of purification and reconstitution should be draw in systematic way.
Section 3.3 adding a table will be more informative.
Section highlight main drugs its components and sources in a table
Overall the study is well design but systematic information such as tables and schemes are missing.
Author Response
see file

Reviewer 2 Report
The manuscript by Cecon et al. reviews relevant information on melatonin receptors in a historical perspective, mainly focusing on receptor discovery, characterization and expression in heterologous cells, on pharmacological tools for receptor characterization and on approved drugs.
This is a nice story recapitulating some on the milestone events in the field of melatonin receptors and ligands, written with the contribution of authors who participated to the discoveries and achievements described. My opinion is that the manuscript should be accepted for publication with only minor modifications and additions.
In particular, I think that in the Introduction the authors should mention and discuss their opinion on nuclear receptors as potential targets of melatonin.
Additionally, the authors should think about the possibility to modify Figure 1, replacing panels C and D with a close-up view of the melatonin binding site. hGPR50 and xMel1c structures are not experimental, but predicted ones, and do not add information to the review.
Table 1: references to the reported pKi values should be added. Please check values, as luzindole appears MT1-selective at the human receptors. The authors should also think about the possibility to introduce a Figure with the formulas of melatonin and compounds reported in Table 1 and to give (little) information on Servier compounds (e.g., are they agonists, partial ago, antagonists)?
Line 125: the authors might briefly comment on the activity status of receptors expressed in Pichia. Are they functionally active?
In chapter 4, I suggest to cite controlled-release melatonin which is an approved drug used for insomnia.
Other minor points and typos:
Line 27: insertion of the iodine atom in position 2 of melatonin not only maintains binding affinity but it also increases potency (as usually happens with lipophilic substituents).
Line 45: delete by
Legend to Figure 1: the color code of panels A and B should be introduced.
Line 68: ref [29, 30], instead of [29, 31]?
Lines 68-70 should be rephrased
Line 72: receptor(s)
Line 79: been
Please check 2-[125I]-iodomelatonin or 2-[125I]-iodoMLT throughout the text.
Line 106: different from
Lines 132-136. The sentence is rather obscure to me. Maybe needs to be rephrased?
Line 141: obtention. French word?
Line 146: why the restriction should be modified at will?
Line 153: is enzymatic correct?
Line 176: pKi 3.8 is very low. See for example pKi 6.58 in ACS Omega 2018, 3, 2865−2882.
Line 178: what does heavily refers to? Melatonin is tritiated (only) on the O-methyl group, at least the Perkin-Elmer’s one.
Line 190: a negligible affinity for MT1
Line 192: have the subtype specific radiolabelled ligands been applied in studies on receptor distribution?
Line 217: I would write “high throughput docking technique”, removing virtual.
Line 259: is ref. 85 properly cited?
Line 304: known
Lines 507-509: different and partially opposing effects are suggested for MT1 and MT2 receptors in sleep regulation. As far as I know, at present this has been proven in rodents.
Line 527: physiological systems?
Author Response
see file

Reviewer 3 Report
This is a very interesting and timely review regarding the molecular characterization and pharmacology of melatonin and its receptors . The authors did a substantial job in reviewing the studies in the literature. Here a few comments to improve the review:
Table 1: please ad a column inicating the pharmacological property of each compound (e.g. agonis, antagonist, etc). Yet, more new ligands with relevant affinity for MT1/2 have been sysnthetized in the last decades. Following a non exaustive list, wich I suggest to include in the table:
Rivara, et al.2007 J. Med. Chem. 2007, 50, 26, 6618–6626.
Radogna F, et al.. 2009;239(1):37-45.
Mor, M., et al. 20021. Bioorganic & Medicinal Chemistry, 9(4), 1045–1057.
Patel, et al. 2020. eLife. 2020; 9: e53779.
3. Melatonin receptor ligands (L-201). At the end of this section, authors should mention the novel light-activatable caged melatonin-compounds. These family of comounds results very promising to study the melatonin effects in a temporally controlled manner in cellular and physiological conditions (Somalo-Barranco, et al. J. Med. Chem. 2022, 65, 16, 11229–11240)
L-274: several works found compelling evidence that MT1 is also linked to the Ca++ signaling pathway via Gq/11 proteins and phospholipase C (PLC) activation. Some examplea of work authors could cite are: Godson, C.;Endocrinology 1997, 138, 397–404.; Brydon, L.;. D1999, 13, 2025–2038; Blumenau, J. Pineal Res 2001, 30, 139–146. )
3.6. Antibodies
458: "(#AMR-031, #AMR-031)". Is a typo instead of "#AMR-032" which correspeonds to the Anti-Melatonin Receptor 1B (MTNR1B) Antibody? please double-check it.
In a very recent work (Posa et al 2022. DOI: 10.1111/jpi.12825), the anti-MT2 Alomone #AMR-032 was validated in the periaqueductal gray (PAG) in IHC/IF and wb experiments. Please add this ref.
Table 3: check some mispelling (MT2, "OK" instead of "KO") and add Posa et al 2022 in the table, section MT2.
Author Response
see file

Round 2
Reviewer 1 Report
Best luck for your future work.
Author Response
Thank you
Reviewer 2 Report
see attached file

Author Response
Hi
Thank you for your suggestions:
1/ the ever-elusive melatonin receptor ..line 138 : "Nuclear" was added, of course
2/ In Table 1, the "2-[125I]-MLT" was replaced by "2-iodoMLT". Actually, on the top of this, number for the mice column were not correct. Fixed.
Thank you for having scrutinized those faulty points.
Reviewer 3 Report
The authors have addressed adequately my comments.
Author Response
Thank you